# The Use of Selective Laser Melting in Mandibular Retrognathia Correction

Andrej Čretnik [1,*] and Anita Fekonja [1,2]

[1] Faculty of Medicine, University of Maribor, Taborska Ulica 8, 2000 Maribor, Slovenia
[2] Healthcare Centre Maribor, Department of Orthodontics, Ulica Talcev 9, 2000 Maribor, Slovenia
* Correspondence: andrej.cretnik@ukc-mb.si

**Abstract:** Digitalization and additive manufacturing offer new possibilities in the manufacturing of individualized medical and dental products. In the paper we present the results of the first 30 consecutive growing patients (15 males and 15 females), with a mean age of 13.69 years (SD = 1.26), who were treated for mandibular retrognathia (skeletal Class II malocclusion), using fixed sagittal guidance (FSG) appliance, individually manufactured by selective laser melting (SLM). Lateral cephalometric radiographs were taken before (T0) and after (T1) treatment and a detailed cephalometric analysis was performed. with a special focus on a time period for malocclusion correction. The analyzed data were compared with the control group (CG; treated with intermaxillary Class II elastics) that was matched for pretreatment age and pretreatment cephalometric measurements. Both methods were effective in the correction of Class II malocclusion, but the time period of correction was significantly shorter (16.03 ± 1.09 months vs. 20.65 ± 4.12 months) with the FSG appliance. After treatment visual skeletal and dentoalveolar effects were achieved, with statistically significant differences measured in mandibular incisors inclination (0.45° in FSG and 2.84° in CG) and distance (−0.61 mm in FSG and 0.13 mm in CG), in mandibular first molar inclination (−1.07° in FSG and 1.18° in CG) and overbite (−3.82 mm in FSG and −2.46 mm in CG), all in favor of FSG appliance. After the final mean treatment time of 16.03 ± 1.09 months, visual skeletal and dentoalveolar effects were achieved, with significant differences in sagittal (SNB angle, SNPg angle, mandibular length (CoGn) and consequently decrease in ANB angle) as well as in vertical (lower anterior facial height (LAFH) and gonial angle) measurements noted, with no reported complications. As the time needed for malocclusion correction was comparable with the reports in the traditional use of the functional appliance and as all the cosmetical and functional changes in all the treated patients remained stable after a 2-year observational period, growing patients with Class II malocclusion could benefit with this type of treatment. As all the cosmetical and functional changes in all the treated patients remained stable after a 2-year observational period, growing patients with Class II malocclusion could benefit from the treatment with FSG appliance.

**Keywords:** selective laser melting; individualized manufacturing technology; dentistry; mandibular retrognathia

## 1. Introduction

Additive Manufacturing (AM) is the term used to describe technologies for creating 3D objects, using a computer, 3D modeling software (Computer Aided Design or CAD), and by depositing material layer by layer, be it solid (plastic, metal, concrete . . . ), liquid, powder or possibly some other materials [1,2]. Additive manufacturing (AM) is defined by ASTM society as a process of joining materials to make parts from 3D model data, usually layer upon layer, as opposed to subtractive manufacturing methodologies [1]. Several types of material (such as polymer, metal, ceramic or composite) can be used for AM [1–3] and intensive research with models about various manufacturing conditions, process parameters, factors and thermophysical properties of the materials are conducted [4–8]. Besides

the other fields in industry and modern life AM offers individually adapted possibilities in medicine (particularly in orthopedics) as well as in dentistry for the production of medical implants, tools, dentures and individual appliances [9–12].

Mandibular retrognathia (Class II malocclusion) is a common orthodontist's challenge as it occurs according to the National Health and Nutrition Estimates Survey III (NHANES III) in 23% of children, 15% of youths and 13% of adults [13]. In most cases, Class II problems are inherited genetically, and may be due to insufficient growth of the mandible, excessive growth of the maxilla or a combination of both [13,14]. A common practice to correct a retrognathic mandible is to use a functional (removable or fixed) appliance, to stimulate mandibular growth in growing patients [15,16] as well as orthognathic surgery with repositioning of the jaw (mandible) in adults [17]. If possible, growth modification is the best way to correct a jaw discrepancy as it allows the grow out of the skeletal disharmony [18,19]. For a good treatment result exact knowledge of growth is necessary and growth patterns must be taken into account when planning treatment [18,19].

In the paper we present the use of fixed sagittal guidance (FSG) appliance, individually manufactured by selective laser melting (SLM) for the treatment of mandibular retrognathia correction in growing patients with horizontal or normal growth patterns. To evaluate the efficiency-effectiveness of this new treatment method, we compared the results with a control group of patients treated with intermaxillary Class II elastics.

## 2. Materials and Methods

The study was reviewed and approved by the Institutional Review Board (No. 01/14) and was conducted in accordance with the declaration of Helsinki at the Orthodontic Department Healthcare Centre. Informed consent approval was obtained from each patient or their parents.

Inclusion criteria were growing patients with bilateral Class II malocclusion (ANB > 4°) due to mandibular retrognathia (SNB < 78°) with overjet (>4 mm) and deep bite (>4 mm), with no previous orthodontic treatment. Patients with developmental syndromes and anomalies (alveolar cleft and/or palate), mandibular fracture or deformities were also excluded from the study. A Class II malocclusion due to retrognathic mandible was identified through cephalographic analysis, so before-treatment (T0) and after-treatment (T1) lateral cephalograms (LC) were mandatory for all the patients. All lateral cephalograms (LC) were taken using the same equipment (Planmeca Promax, Helsinki, Finland) by an experienced dental radiology engineer under standard conditions: subjects were in the standing position and adequately protected, the Frankfort horizontal plane parallel to the floor, with the teeth in the maximal intercuspation (centric occlusion) and relaxed lips and tongue and with identical distances for each patient from the focus to the median sagittal plane of the subject's head and to the film. Cephalometric analysis was performed for each patient before (T0) and after (T1) treatment. Definitions of the used (clinically important) angular and linear measurements are described and shown in Figures 1 and 2. The linear and angular measurements were measured to the nearest of 0.1 mm and 0.1 degrees, respectively. Only the data with statistically significant changes before and after treatment are presented.

Legends: SNB: the angle formed between the SN plane and the point B indicates the relationship of the mandibular basal arch to the anterior cranial base; SNPg: the angle formed between the SN plane and the point Pg; ANB: the difference between SNA and SNB angle relates jaws to anterior cranial base; Wits appraisal: linear distance between the projecting points A and B perpendicular on the functional occlusal plane (AO and BO) indicates the skeletal sagittal jaw relationship; Gonial angle: the angle between the posterior tangent line of the ramus and the mandibular plane; LAFH: Lower anterior face height is the distance between points Sna and Me; CoGn: the linear distance between the condylion and the gnathion points indicates the mandibular length.

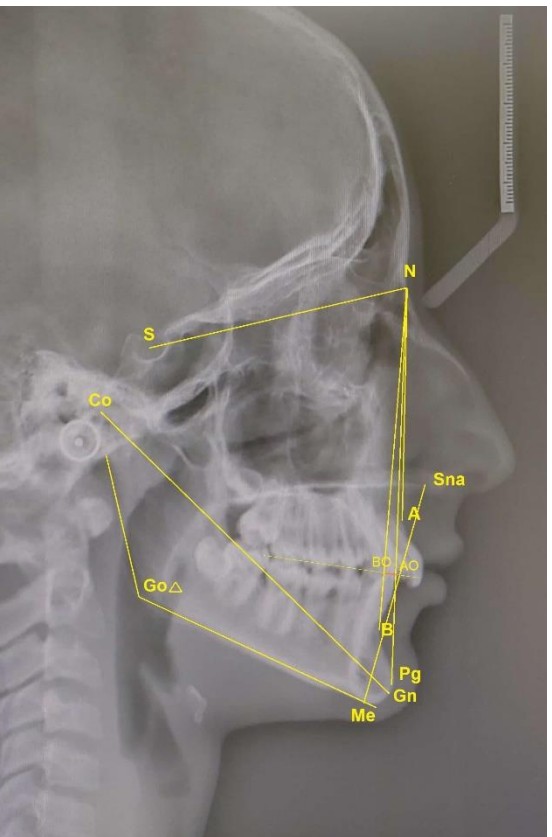

**Figure 1.** The angular and linear skeletal measurements.

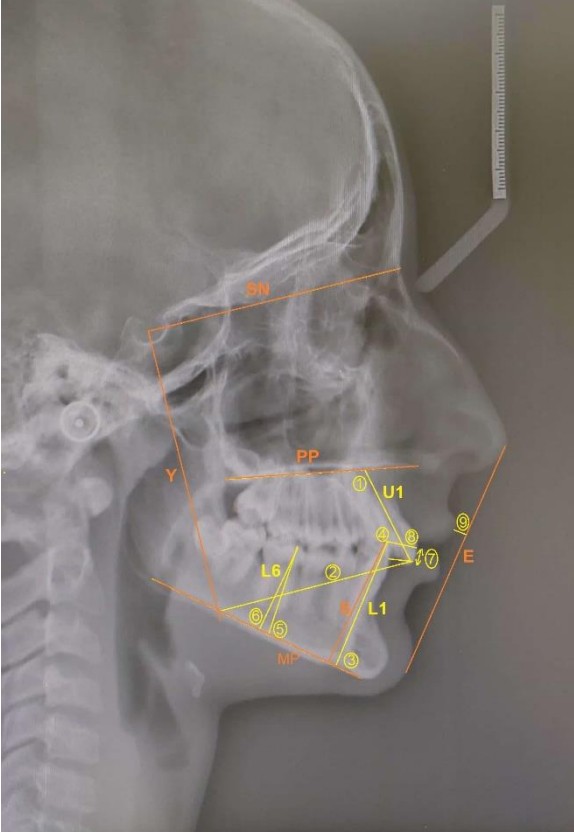

**Figure 2.** The angular and linear dentoalveolar measurements.

Legends: 1: U1/PP (°) is the angle formed between the maxillary central incisor (U1) long axis and the palatal plane (PP) and indicates maxillary incisor inclination; 2: U1/Y axis (mm) is the distance between the incisal edge of maxillary central incisor perpendicular to the Y axis (formed by dropping a line from the sella perpendicular to the SN line) indicates horizontal maxillary incisor distance; 3: L1/MP (°) is the angle formed between the mandibular central incisor (L1) long axis and the mandible plane (MP) and indicates mandibular incisor inclination; 4: L1/S axis (mm) is the distance between the incisal edge of the mandibular central incisor perpendicular to the S axis (formed by dropping a line through the posterior conture of symphysis perpendicular to the MP line) and indicates horizontal mandibular incisor distance to S axis; 5: L6/MP (°) is the angle between the mandibular first molar (L6) long axis (line passing through the mesial cusp tip and the mesial root tip) and the mandibular plane, and indicates mandibular first molar inclination; 6: L6/MP (mm) is the distance between the mesial cusp tip of the mandibular first molar perpendicular to the mandibular plane and indicates vertical mandibular first molar distance; 7: overjet is the horizontal distance from the maxillary incisor tip to the labial surface of the mandibular incisor; 8: overbite is the vertical distance from the mandibular incisor tip to the maxillary incisor tip; 9: U1/E line is the horizontal distance from the labial surface of the maxillary incisor perpendicular to the E line.

The fixed sagittal guidance (FSG) appliance is a fixed orthodontic appliance bonded on both upper molars, manufactured of a crown (cobalt-chromium alloy) and the occlusal inclined plane of SR Chromasit material (pressure/heat-curing micro filled veneer material) (Figure 3). Crowns (Figure 4) were produced using a scanner (to import physical data about the teeth into the computer) and 3D computer design and a specific method of selective laser melting (SLM) (MLab, Concept Laser, Treatstock, Newark, DE, USA) manufacturing technology (adding a layer on top of the layer) [5]. Special small hooks were added to the crowns for bondage and safety of the appliance (Figure 4). The inclination was oriented individually to the occlusal plane (angle), thereby actively guiding the mandible anteriorly during jaw closure; it was individually manufactured in a laboratory using an articulator. A construction wax bite was necessary in designing an inclined plane. In order to register the bite for FSG manufacturing, the patient was asked to close his/her mouth in proper sagittal and vertical dimensions. The inclination was adjusted in the laboratory on a case-to-case bias (depending on the severity of the Class II relationship, deep bite) [11]. The vertical distance of the inclined plane was temporarily reflected in a posterior open bite which enabled easier leveling of the mandibular teeth and correction (flattening) of the curve of Spee. Additionally, all the patients were treated with the same 0.22″ slot Roth prescription brackets (Dentaurum) and consistent straight (arch) wire sequencing (SWA) (Figure 3). In leveling and aligning, the arch wire sequence was 0.012, 0.014, 0.018, 0.016 × 0.022, 0.018 × 0. 025 inch rectangular nickel-titanium wire. The FSG was placed at the same time as the fixed orthodontic straight-wire appliance (SWA) (Figure 3).

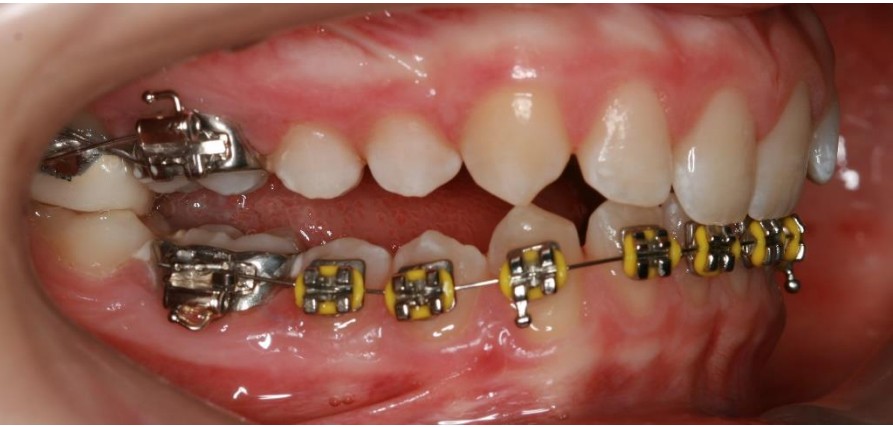

**Figure 3.** The fixed sagittal guidance (FSG) and fixed orthodontic (SWA) appliance.

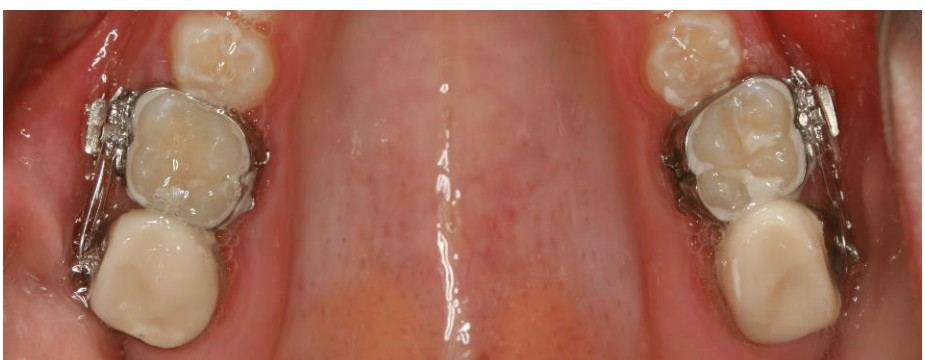

**Figure 4.** Crowns (with safety hooks) were produced using 3D computer design and additive manufacturing technology (selective laser melting (SLM)).

Final skeletal and dentoalveolar results of the patients treated with FSG were compared with the same results of patients treated with intermaxillary Class II elastics (control group), who matched to the FSG group in age and initial cephalometric characteristics.

*Statistical Analysis*

Intra-operator error was evaluated by re-digitizing fifteen randomly selected cephalograms two weeks after initial digitization. Error analysis was performed using paired *t*-tests. Descriptive statistics (mean and standard deviation) for measurements before and after treatment were analyzed using the Statistical Package for Social Sciences version 10.0 (SPSS Inc., Chicago, IL, USA).

An independent *t*-test was used to compare cephalometric measurements before and after treatment within each group and paired *t*-test for comparison between groups. The level of significance tested was $p < 0.05$.

## 3. Results

We prospectively analyzed the results of the first 30 consecutive patients (15 males and 15 females) with a mean age of 13.69 years (standard deviation [SD] $\pm$ 1.26 years), who accepted to be treated with FSG in conjunction with SWA, with a control group of the last 30 consecutive patients (14 males and 16 females) with the mean age of 13.82 years (standard deviation [SD] $\pm$ 1.09 years) treated with intermaxillary Class II elastics. There were no statistically significant differences found between groups in the initial age of treatment and in gender distribution ($p > 0.05$)

Intra-operator error analysis showed no statistically significant difference observed in the cephalometric analyses between groups (NS, $p > 0.05$). The correlation values did not reveal any systematic measurement error (correlation coefficients for linear and angular values were 0.85 and 0.89, respectively).

The mean treatment time of 16.03 $\pm$ 1.09 months in the FSG group was found to be statistically significantly shorter in comparison to 20.45 $\pm$ 4.12 months in the elastic group ($p < 0.05$).

Cephalometric measurements of sagittal and vertical variables with statistically significant differences achieved before (T0) and after (T1) treatment in both groups, are presented in Table 1.

**Table 1.** The mean changes within each group and comparison between both groups.

| Cephalometric Variable | FSG Group Mean Change $\pm$ SD | Test | Elastic Group Mean Change $\pm$ SD | Test | Group Difference |
|---|---|---|---|---|---|
| SNA (°) | 0.46 $\pm$ 0.47 | NS | 0.35 $\pm$ 0.73 | NS | NS |
| SNB (°) | 2.68 $\pm$ 1.84 | <0.05 | 2.54 $\pm$ 1.96 | <0.05 | NS |
| SNPg (°) | 2.71 $\pm$ 1.46 | <0.05 | 2.66 $\pm$ 1.28 | <0.05 | NS |

**Table 1.** *Cont.*

| Cephalometric Variable | FSG Group Mean Change ± SD | Test | Elastic Group Mean Change ± SD | Test | Group Difference |
|---|---|---|---|---|---|
| ANB (°) | −2.14 ± 0.84 | <0.05 | −2.09 ± 0.68 | <0.05 | NS |
| Wits (mm) | −2.54 ± 1.03 | <0.05 | −2.02 ± 1.63 | <0.05 | NS |
| Gonial angle | 2.34 ± 1.37 | <0.05 | 1.66 ± 1.23 | <0.05 | NS |
| LAFH (mm) | 2.65 ± 0.93 | <0.05 | 2.18 ± 0.64 | <0.05 | NS |
| CoGn (mm) | 3.59 ± 0.89 | <0.05 | 2.86 ± 1.02 | <0.05 | NS |
| U1/PP (°) | −6.28 ± 2.31 | <0.05 | −5.95 ± 2.89 | <0.05 | NS |
| U1/Y axis (mm) | −3.40 ± 0.15 | <0.05 | −2.71 ± 0.44 | <0.05 | NS |
| L1/MP (°) | 0.45 ± 2.71 | NS | 2.84 ± 3.32 | <0.05 | <0.05 |
| L1/S axis (mm) | −0.61 ± 0.48 | <0.05 | 0.13 ± 0.80 | NS | <0.05 |
| L6/MP (°) | −1.07 ± 1.22 | <0.05 | 1.18 ± 1.89 | <0.05 | <0.05 |
| L6/MP (mm) | 2.41 ± 0.66 | <0.05 | 2.56 ± 0.67 | <0.05 | NS |
| Overbite (mm) | −3.82 ± 1.40 | <0.05 | −2.46 ± 1.57 | <0.05 | <0.05 |
| Overjet (mm) | −3.69 ± 1.84 | <0.05 | −3.68 ± 1.33 | <0.05 | NS |
| U lip to E plane (mm) | −1.35 ± 0.46 | <0.05 | −1.54 ± 0.80 | <0.05 | NS |

There were statistically significant changes found within both groups in all the parameters except in SNA (°), L1/MP (°) and L1/S axis (mm).

We found statistically significant differences between groups in dentoalveolar measurements of the mandibular incisor (L1/MP—proclined in both groups (0.45 ± 2.71° in FSG and 2.84 ± 3.32° in CG) and in L1/S axis (moved distally for 0.61 ± 0.48 mm in FSG and mesially for 0.13 ± 0.80 mm in CG). Mandibular molar significantly extruded in FSG for 2.41 mm and in EG for 2.56 mm but difference between groups was not statistically significant. The mandibular molar (L6/MP) significantly inclined mesially (1.18° ± 1.89) in CG while in FSG had a more upright position (−1.07° ± 1.22), with a statistically significant difference between groups.

Visual skeletal and dentoalveolar effects were achieved in both groups (Figure 5), with statistically significant differences in sagittal (SNB angle, SNPg angle, mandibular length (CoGn) and consequently decrease in ANB angle and Wits) as well as in vertical (lower anterior facial height (LAFH) and gonial angle) measurements, but with no statistically significant differences between groups.

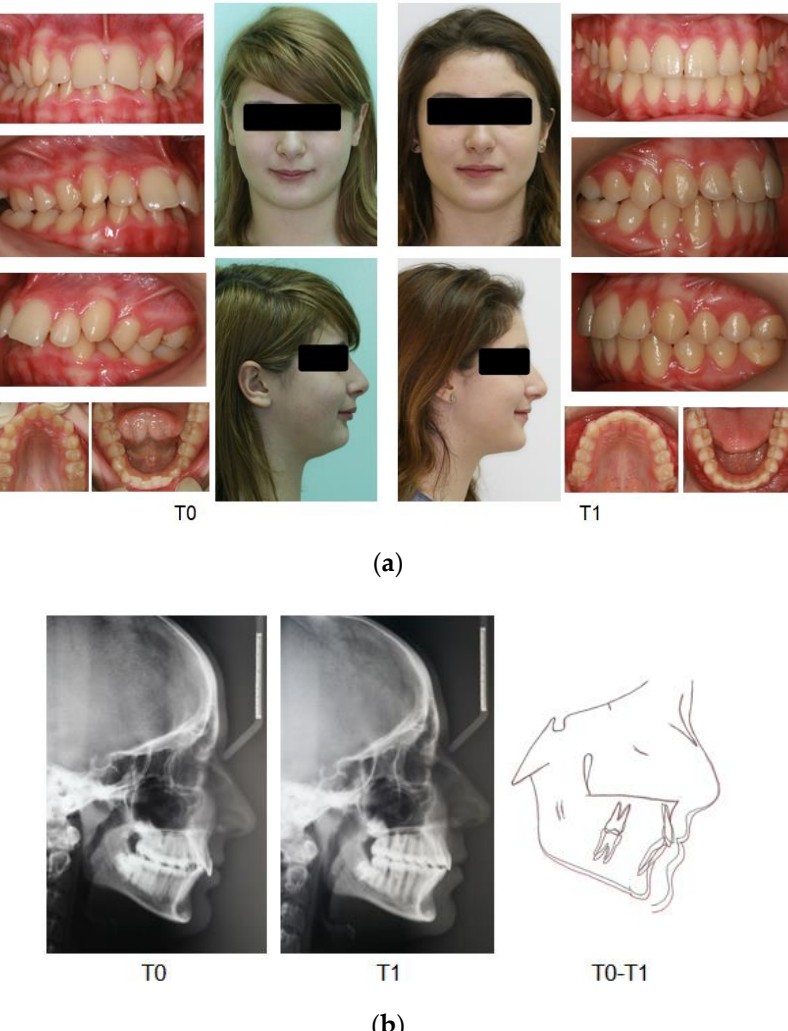

**Figure 5.** (**a**) Intraoral and extraoral photos before (T0) and after (T1) treatment with FSG (SWA). (**b**) Lateral cephalograms before (T0) and after (T1) treatment, and superimposition (T0-T1).

## 4. Discussion

Digitalization and additive manufacturing have opened many new perspectives in the field of personalized complex medical and dental implant production [2,3,9–12,20,21]. Individualized approach and custom-made implants offer attractive and seem to be almost unlimited options. On the other side we must be aware of concerns about quality and safety assurance control and regulation protocols, particularly in the very sensitive field of medical implants. As these issues seem to play a slightly less important role in temporarily and externally worn implants, the highest ethical and deontological standards should be respected, particularly in permanently inserted and implanted medical devices.

In orthodontics, many corrections can be achieved through the growing and maturation periods and with externally applied devices [13–16]. There's a clear trend, if possible, to achieve similar orthodontic results, to perform this with more convenient and effective methods for patients [21,22]. Removable appliances, such as the Fränkel regulator, Bionator, Activator, Twin Block and Class II elastics are effective in the treatment of malocclusions, but often bring inconsistent results, due to the fact that they require high levels of patient's cooperation [23–26]. The major advantage of fixed functional devices such as the Herbst, Jasper Jumper, MARA and Forsus Fatigue Resistant Device lies in their fixed and effective position for 24 h a day with a low impact of patient's compliance [23–27].

The FSG appliance, used in our study, was completely individually designed and fitted perfectly on the upper molars, so no additional correction was needed at the time of bonding; it provided guidance of mandible anteriorly and inferiorly, with the impact in the correction of the sagittal and vertical discrepancy in each closing of the jaw and what is supposed to induce a neuromuscular re-education, while correcting the Class II dentoskeletal relationship [24]; this approach also prevents the teeth from occluding with antagonists and allows immediate use of a fixed orthodontic appliance in the lower arch, in contrast to for instance Class II elastics, that can be used only after completed alignment of teeth. FSG follows the principle of a bionator, which facilitates the eruption of mandibular posterior teeth by trimming of the acrylic [23]. In FSG, temporarily achieved posterior teeth space (open bite) with leveling of the mandibular teeth and correction (flattening) of the curve of Spee with avoiding downward and back rotation of mandible, is provided by (individually) inclined plane on FSG.

The above-mentioned concept of treatment also seems to play a role in the reduction of the mean correction time with the appliances. Nelson et al. reported the mean correction time of dental relationship with Class II elastics to be between 6 months and 1.3 years [28]. Uzel et al. reported about a reduction in the mean correction time of $8.5 \pm 2.6$ months using Class II elastics comparing with the mean correction time of $4.6 \pm 1.7$ months, using the fixed Reciprocal Mini-Chin Cup (RMCC) appliance [29]. The mean correction time in our patients, treated with the proposed FSG appliance, was not as short as in the report from Uzel et al. [29] but comparable to that of Nelson et al. [28] and there could be several reasons for that, including maturation stage (growing potential) of the patients with their cooperation and possible ability of neuromuscular re-education [24,27,28,30]; it should be stressed but that the proposed FSG appliance enabled immediate use of SWA, without waiting period, such as with the use of Class II elastics, that could be used only after completed alignment of teeth.

Effective treatment of Class II malocclusions should generate the skeletal (orthopedic) and dentoalveolar (orthodontic) effects; this concept with aligning of teeth and correction of malocclusion has been confirmed in the results of skeletal and dentoalveolar measurements in our study, too. According to several authors, Herbst appliance, Jasper Jumper, Forsus, FMA and intermaxillary elastics have a tendency to procline the mandibular incisors and effect on mandibular molar mesialisation [26,31,32]. We found the mandibular incisor and molar in the FSG group to be statistically significant less proclined as in the control group. As the FSG has no influence on mandibular incisors proclination, such as elastics have, this appliance might be ideal in the situation of proclined mandibular incisors, where there is a great need to control incisor inclination. Similar results with retrusion of the mandibular incisor were reported by Ozbilek et al. [33] and Celikoglu et al. [34].

As the correction time in our study took more than a year, several factors in treatment should also be considered, such as the safety of the treatment and appliance, failure rate, patients' compliance and the long-term results. With the proposed construction and bonding, there were no complications observed in the treatment of our patients, who all successfully finished the same protocol of treatment. Within the follow-up period of 2 years, there were no observed changes and no clinical, nor subjective worsening of the achieved results after finishing of the treatment.

## 5. Conclusions

In the results of the presented study, we found individually manufactured fixed sagittal guidance (FSG), combined with fixed orthodontic (SWA) appliance, as the safe and effective method in correcting Class II malocclusion due to mandibular retrognathia in growing patients.

The results of our study confirmed that individually manufactured fixed sagittal guidance (FSG) appliance and intermaxillary Class II elastics were both effective in correction of Class II malocclusion in our patients, but the average period of time needed for correction

was statistically significantly shorter (for 4.39 months) with FSG appliance in comparison to Class II elastics ($p < 0.05$).

As the time needed for malocclusion correction was comparable with the reported time in the traditional use of the functional appliance and as all the cosmetical and functional changes in treated patients remained stable after 2 years' observational period, growing patients with Class II malocclusion could benefit with this type of treatment.

In the results of the presented study, we found individually manufactured FSG, combined with a fixed orthodontic (SWA) appliance, as the safe and effective method in correcting Class II malocclusion due to mandibular retrognathia in growing patients, who might benefit from this type of treatment.

**Author Contributions:** Conceptualization, methodology, investigation, data analyses, writing—original draft preparation, writing—review and editing, analysis, A.Č. and A.F. All authors have read and agreed to the published version of the manuscript.

**Funding:** This research received no external funding.

**Institutional Review Board Statement:** Approved by the Institutional Review Board (No. 01/14).

**Informed Consent Statement:** Informed consent was obtained from all patients' parents or patients included in the study.

**Data Availability Statement:** The data presented in this study are available on request from the corresponding author.

**Conflicts of Interest:** The authors declare no conflict of interest.

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
