# Peer review of "The Use of Selective Laser Melting in Mandibular Retrognathia Correction"

_metals, doi:10.3390/met12091544_

Round 1

Reviewer 1 Report

Dear Authors, 

I think that the article is well written but important records must be added.

-Firstly, all photographic records are required, for example full occlusal photos, sagittal photos (left and right side) and all extra oral photos. 

-A final lateral cephalogram is important to compare skeletal improvements, together with the superimposition of lateral cephalogram at T0 and T1

-Regarding the inclusion criteria I also think you should consider the Witz index (AO-BO).

-In the introduction, you should better explain when a mandibular advancement should be done in growing patient. Indeed, a mandibular advancement is not indicated in all cases of retrognathic mandible but in particular when the ramus is smaller, this to stimulate the growth of it. In view of this how your fixed appliance can stimulate the growth?

- Another question is: How do you manage the temporary posterior open bite at the beginning of the treatment? the other functional appliances usually have a sort of shield that stops the tongue in its lateral movements, but in your case, is there a risk that the tongue can obstacle extrusion movements during the first phases of treatments? 

- Are there some limitations in this study?

- Was the FSG left over the all orthodontic treatment? When it was bonded, was the second upper molar totally erupted? please better explain all of this. 

Author Response

Please find enclosed: Answer to Reviewer 1

Reviewer 2 Report

This paper reports the individually manufactured fixed sagittal guidance made by SLM can be used for treatments of Class 2 malocclusion. However, there is no outcome that shows the efficiency of this new method, because there is no control group in this study. If authors want to prove the efficiency of new method, the control group must be set and some outcome, such as treatment time for bonding and cost or questionnaire survey, must be compared with a traditional method.

However, if how to design and make the FSG was described more precisely, this paper could be acceptable for a case report for orthodontic journals or clinical dental journals. The submission to other dental journal as a case report is highly recommended.

Author Response

Please find enclosed Answers to Reviewer 2

Reviewer 3 Report

The manuscript entitled “metals-1818140-LB-PBF” dealing with AM has been reviewed. The paper has been nicely written but needs significant improvement. Please follow my comments.

1.     Please follow the ASTM 52900 for correct terminologies in AM.

2.     Provide more discussion and fundamental relations for figure 2 “The angular and linear dentoalveolar….

3.     Add some quantitative results to the abstract.

4.     What is the future direction of this work?

5.     Laser absorptivity in AM is important which shows the quality of the parts and transition from keyhole to conduction mode. Please read and add the following ref in this area. “The effect of absorption ratio on meltpool features in laser-based powder bed fusion of IN718”.

6.     Please update the introduction with the new publications in the field. Authors are encouraged to read and add the following new papers in the field.

·        High-cycle fatigue properties of curved-surface AlSi10Mg parts fabricated by powder bed fusion additive manufacturing

·        Proposal of design rules for improving the accuracy of selective laser melting (SLM) manufacturing using benchmarks parts

·        Fatigue life optimization for 17-4Ph steel produced by selective laser melting

·        Evolution of temperature and residual stress behavior in selective laser melting of 316L stainless steel across a cooling channel

Author Response

Please find enclosed Answers to Reviewer 3

Round 2

Reviewer 2 Report

The revised manuscript clearly shows the efficiency of using new method to shorten treatment time.  However study design, like prospective or retrospective, is not shown. Authors should describe the study design more precisely and how the patients were divided into each group.

Reviewer 3 Report

The paper is ready to publish.

Author Response

See Comments - The paper is ready to publish.